# Incidence and treatment trends of infectious spondylodiscitis in South Korea: A nationwide population-based study

**Hee Jung Son[1], Myongwhan Kim[1], Dong Hong Kim[2], Chang-Nam Kang[2]\***

**1** Department of Orthopedic Surgery, Nowon Eulji Medical Center, Eulji University School of Medicine, Seoul, South Korea, **2** Department of Orthopedic Surgery, Hanyang University College of Medicine, Seoul, South Korea

\* cnkang65@hanyang.ac.kr

**Data Availability Statement:** Data are available from the Korea National Health Insurance Sharing Service (https://nhiss.nhis.or.kr/bd/ay/bdaya001iv.do;jsessionid=

## Abstract

The incidence of infectious spondylodiscitis (IS) has increased in recent years due to an increase in the numbers of older patients with chronic diseases, as well as patients with immunocompromise, steroid use, drug abuse, invasive spinal procedures, and spinal surgeries. However, research focusing on IS in the general population is lacking. This study investigated the incidence and treatment trends of IS in South Korea using data obtained from the Health Insurance Review and Assessment Service. A total of 169,244 patients (mean age: 58.0 years) diagnosed from 2010 to 2019 were included in the study. A total of 10,991 cases were reported in 2010 and 18,533 cases in 2019. Hence, there was a 1.5-fold increase in incidence rate per 100,000 people from 22.90 in 2010 to 35.79 in 2019 ($P <$ 0.05). The incidence rate of pyogenic spondylodiscitis per 100,000 people increased from 15.35 in 2010 to 33.75 in 2019, and that of tuberculous spondylodiscitis decreased from 7.55 in 2010 to 2.04 in 2019 ($P <$ 0.05, respectively). Elderly individuals $\geq$ 60 years of age accounted for 47.6% (80,578 patients) of all cases of IS. The proportion of patients who received conservative treatment increased from 82.4% in 2010 to 85.8% in 2019, while that of patients receiving surgical treatment decreased from 17.6% to 14.2% ($P <$ 0.05, respectively). Among surgical treatments, the proportions of corpectomy and anterior fusion declined, while proportion of incision and drainage increased ($P <$ 0.05, respectively). The total healthcare costs increased 2.9-fold from \$29,821,391.65 in 2010 to \$86,815,775.81 in 2019 with a significant increase in the ratio to gross domestic product. Hence, this population-based cohort study demonstrated that the incidence rate of IS has increased in South Korea. The conservative treatment has increased, while the surgical treatment has decreased. The socioeconomic burden of IS has increased rapidly.

## Introduction

Infectious spondylodiscitis (IS) is an infection of the vertebral body, intervertebral discs, nervous tissue, and surrounding soft tissues. IS is a relatively rare disease, accounting for 3–5% of

total osteomyelitis cases. The incidence rate of IS is 2.2–5.8 per 100,000 people per year. The incidence is higher in older adults, at 9.8 per 100,000 people ≥ 65 years of age [1–4]. However, 27–32% of patients diagnosed with IS develop serious complications such as muscle weakness or paralysis, impaired urinary or bowel symptoms, persistent back pain, and spinal deformity. In addition, IS can be a life-threatening disease, with reported mortality rates of 2–29% [5–7].

The most crucial factor in establishing a diagnosis of IS is a high clinical suspicion of infection based on symptoms such as back pain and fever. White blood cell count, C-reactive protein, and erythrocyte sedimentation rate should be assessed in cases with a high clinical suspicion of IS. In addition, a blood culture should be performed to confirm bacteremia. These patients should also undergo magnetic resonance imaging using a contrast media, which is the most accurate test for diagnosing IS, as well as psoas and epidural abscesses [3, 8–10].

The conservative treatment for IS includes the administration of antibiotics or anti-tuberculosis drugs, as well as bed rest, and spinal stabilization with orthosis. Surgical treatment such as incision and drainage, decompression, corpectomy, and fusion can also be required in certain cases. Surgery is usually indicated for bacteriological or histological confirmation, presence of intractable pain, neurological paralysis, or spinal deformity. It can also be performed if there is lack of improvement in symptoms despite the administration of appropriate antibiotics or anti-tuberculosis drugs. However, IS is often difficult to diagnose early, with an average of 3 months required to reach the diagnosis. Its treatment also often requires an average of ≥ 12 months for completion of therapeutic regimen. This can pose significant difficulty in managing this disease appropriately. It is also associated with a significant burden, both individually and socioeconomically [3, 11–13].

In recent years, the incidence of IS, especially pyogenic spondylodiscitis (PS), has been increasing due to an increase in the numbers of older patients with chronic diseases, immunocompromised patients, as well as patients with a prior history of steroid use, drug abuse, invasive spinal procedures, and spinal surgeries. Early diagnosis due to the development of imaging tests has also increased the incidence of IS. However, the incidence of tuberculous spondylodiscitis (TS) has reportedly decreased with the decrease in the incidence of tuberculosis [7–9, 14, 15]. The awareness of increasing IS incidence can lead to its early detection. It would also help clinicians avoid missed diagnosis. However, research focusing on IS in the general population is lacking. Hence, this study investigated the incidence and treatment trends of IS in South Korean population using data obtained from the Health Insurance Review and Assessment Service (HIRA).

## Methods

### Data source

This study used data recorded in the HIRA database from 2010 to 2019. The South Korean government has adopted a form of social insurance that provides medical benefits through the National Health Insurance (NHI). Cosmetic procedures, and medical care required after traffic or occupational accidents are excluded from coverage under this insurance. In total, 97% of the South Korean population is enrolled in the NHI, while the 3% with very low incomes are classified as eligible for the Medical Aid Program (MAP). HIRA manages all medical data of patients enrolled in the NHI and MAP. Therefore, the HIRA database contains a record of all medical care services provided to almost all the patients in Korea. Database information includes diagnoses based on the International Classification of Diseases, 10th revision (ICD-10), demographic data such as age and sex, prescribed medications, tests, procedures, and surgeries performed. These data are suitable for studies investigating the incidence rate or treatment trends of a particular disease in the entire population.

## Study design and cohort

IS can be broadly classified into two types; pyogenic and granulomatous spondylodiscitis. The causes of PS are mostly bacterial, while granulomatous spondylodiscitis can be caused by Mycobacterium tuberculosis, and certain other bacteria and fungi, etc. As most cases of granulomatous spondylodiscitis are caused by Mycobacterium tuberculosis, the present study focused on PS and TS. These conditions were classified in the HIRA database by ICD-10 codes M462 (osteomyelitis of vertebra), M463 [infection of intervertebral disc (pyogenic)], M465 (other infective spondylopathies), M492 (enterobacterial spondylitis), M493 (spondylopathy in other infectious and parasitic diseases), M490 and A1800 (tuberculosis of the spine). In order to improve the accuracy of the diagnosis, IS cases were defined as patients assigned the above diagnosis codes who visited a medical institution three or more times with the same diagnosis codes assigned for these visits. This study included a total of 169,244 patients (mean age: 58.0 years) who were diagnosed in the last 10 years. Patients diagnosed with IS prior to this period were excluded.

Patients who underwent surgery for IS after the diagnostic code was recorded were classified as having received surgical treatment. The remaining patients were classified as having received conservative treatment, including antibiotics or anti-tuberculosis drugs. The surgical treatment methods were classified using the following codes: N1491-3, N1497-8, N2491, N2497-9 (decompression), N0451-3 (corpectomy), N2461-6, N0466, N1466 (anterior fusion), N2467-9, N0468-9, N1460, N1469, N2470 (posterior fusion). Patients who were not assigned the aforementioned surgical codes but were confirmed to have undergone surgery were considered to have received incision and drainage.

The demographic data and comorbidities of patients diagnosed with IS based on the ICD-10 code were also examined. The comorbidities were assessed using the following ICD-10 codes: E10-4 (diabetes mellitus, DM), N18-9 (end-stage renal disease, ESRD), M05-6, M08 (rheumatoid arthritis, RA), D86, D89, M30-6 (autoimmune disease except for RA), K74 (liver cirrhosis), J44 (chronic obstructive pulmonary disease), B20, D80-5, D87-8 (immunodeficiency disease), I33, and I38-9 (infective endocarditis). The comorbidities were confirmed when there were three or more visits to a medical institution with the same diagnosis codes recorded at each visit, and when the prescription of drugs suitable for the disease was also reviewed.

Additionally, the proportion of IS patients with NHI and MAP enrollment were reviewed and the total amount claimed from HIRA was identified. The claimed amounts were identified in Korean won and converted to US dollars using the average exchange rate (1$ = 1,166₩) in 2019 (http//ecos.bok.or.kr). The ratio of total healthcare costs to gross domestic product (GDP) were subsequently calculated to determine the socioeconomic burden.

## Ethical statement

This study was approved by the Institutional Review Board (IRB No. 2023-01-025). Due to the retrospective nature of this study, and use of data stored in the HIRA's database, the requirement for patient consent was waived. And all data obtained from HIRA's database were fully anonymized.

## Statistical analysis

The Cochran-Armitage trend test was performed to analyze the incidence of IS and the treatment trends over 10 years. The changes in the ratio of total healthcare costs to GDP were analyzed using chi-square tests with linear-by-linear association. The proportions of NHI enrollees and MAP subjects and the ratio of men and women patients were compared using

chi-square tests. All statistics were performed using SAS version 9.3 (SAS Institute Inc., Cary, NC, USA). $P < 0.05$ was considered statistically significant.

## Results

The incidence rate of IS in South Korea during a 10-year-period from 2010 to 2019 was 33.4 per 100,000 people. A total of 10,991 cases were reported in 2010 and 18,533 cases in 2019. There was a 1.5-fold increase in the IS incidence rate per 100,000 people, from 22.90 in 2010 to 35.79 in 2019 ($P < 0.05$). The PS incidence rate per 100,000 people increased from 15.35 in 2010 (7,368 patients) to 33.75 in 2019 (17,477 patients), while the TS incidence rate per 100,000 people decreased from 7.55 (3,623 patients) in 2010 to 2.04 (1,056 patients) in 2019 ($P < 0.05$, respectively) (Table 1 and Fig 1). The highest number of patients were in their 50s (37,132 patients). The elderly population $\geq 60$ years of age accounted for 47.6% (80,578 patients) of the total cases of IS (Table 2). The incidence in both men and women increased from 2010 to 2019, with a higher incidence observed in women than in men ($P < 0.05$, respectively) (Fig 2).

The proportion of patients who received conservative treatment increased from 82.4% in 2010 to 85.8% in 2019. In contrast, the number of patients who underwent surgical treatment decreased from 17.6% to 14.2% ($P < 0.05$, respectively). Among the different surgical treatments, the proportions of corpectomy and anterior fusion declined, while proportion of incision and drainage increased ($P < 0.05$, respectively) (Table 3). Patients with IS experienced multiple comorbidities, including DM (93,199 patients, 55.1%), RA (46,146 patients, 27.3%), chronic obstructive pulmonary disease (25,742 patients, 15.2%), and ESRD (21,682 patients, 12.8%). Infective endocarditis was identified in 1,133 patients (0.7%), which was mainly associated with PS; therefore, the ratio was calculated based on PS cases (153,333 patients in total) (Table 4).

A significantly higher number of IS cases occurred in patients enrolled in MAP as compared with NHI enrollees ($P < 0.05$) (Table 5). The total healthcare costs increased 2.9-fold from 34,771,742,660 Korean Won ($29,821,391.65) in 2010 to 101,227,194,590 Korean Won ($86,815,775.81) in 2019. The socioeconomic burden of the disease also demonstrated rapid growth due to a significant increase in the ratio to GDP ($P < 0.05$). The medical costs did not differ significantly between patients who received conservative and surgical treatments (Fig 3).

**Table 1. The number of IS cases and its incidence rate per 100,000 people from 2010 to 2019.**

| Year | No. of cases | | | Incidence rate per 100,000 population | | |
|---|---|---|---|---|---|---|
| | Total | Pyogenic | Tuberculosis | Total | Pyogenic | Tuberculosis |
| 2010 | 10,991 | 7,368 | 3,623 | 22.90 | 15.35 | 7.55 |
| 2011 | 13,211 | 11,560 | 1,651 | 26.04 | 22.79 | 3.25 |
| 2012 | 15,793 | 14,054 | 1,739 | 31.00 | 27.58 | 3.41 |
| 2013 | 16,505 | 14,951 | 1,554 | 32.27 | 29.23 | 3.04 |
| 2014 | 19,018 | 17,627 | 1,391 | 37.05 | 34.34 | 2.71 |
| 2015 | 17,959 | 16,576 | 1,383 | 35.17 | 32.46 | 2.71 |
| 2016 | 19,284 | 18,013 | 1,271 | 37.61 | 35.13 | 2.48 |
| 2017 | 19,532 | 18,321 | 1,211 | 37.98 | 35.63 | 2.36 |
| 2018 | 18,418 | 17,386 | 1,032 | 35.67 | 33.67 | 2.00 |
| 2019 | 18,533 | 17,477 | 1,056 | 35.79 | 33.75 | 2.04 |
| Total | 169,244 | 153,333 | 15,911 | | | |
| P-value | < 0.05 | < 0.05 | < 0.05 | < 0.05 | < 0.05 | < 0.05 |

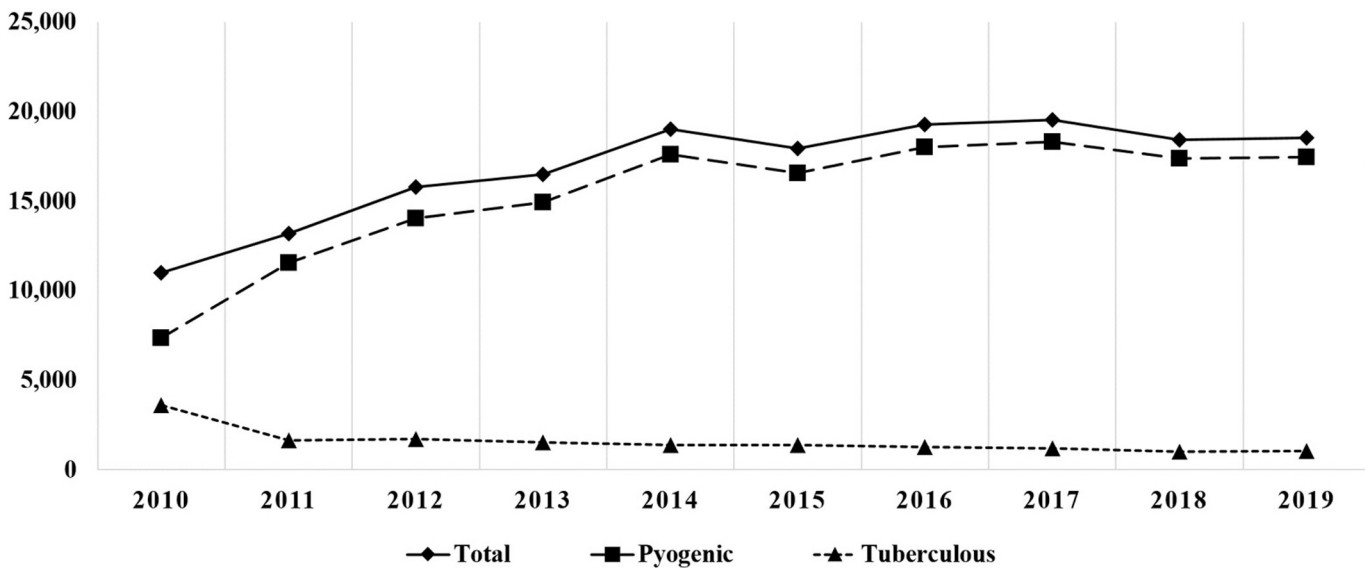

**Fig 1. The number of PS and TS cases from 2010 to 2019.**

## Discussion

The findings of this study demonstrate that the incidence rate of IS per 100,000 people in South Korea has increased 1.5-fold over the past 10 years; the incidence rate of PS has increased while that of TS has decreased. The proportion of patients receiving conservative treatments increased during this period, while that of patients who underwent corpectomy and anterior fusion decreased. The socioeconomic burden of IS has increased rapidly.

Since IS is a relatively rare disease, published reports of its incidence are lacking. Grammatico et al. [16] reported that 1,422 and 1,425 patients with IS were identified in national hospitals in France in the years 2002 and 2003, respectively, with an incidence rate of 2.4 per 100,000 people and 6.5 per 100,000 in people $\geq$ 70 years of age. Kehrer et al. [1] conducted a study in the Danish Funen County that included 483,123 individuals. They reported that the incidence of IS increased from 2.2 in 1995 to 5.8 in 2008. Noh et al. [14] also investigated the incidence rate of IS using data obtained from the HIRA. They reported that the rate of IS increased from 23.3 per 100,000 people in 2005 to 58.4 per 100,000 in 2015. Using a similar

**Table 2. The number of IS cases stratified according to 10-year age groups.**

| Age (years) | No. of cases [mean age: 58.0 (1–105)] | | |
|---|---|---|---|
| | Total | Men | Women |
| $\leq$ 10 | 283 | 155 | 128 |
| 11 ~ 20 | 2,981 | 1,701 | 1,280 |
| 21 ~ 30 | 8,033 | 4,540 | 3,493 |
| 31 ~ 40 | 15,474 | 8,575 | 6,899 |
| 41 ~ 50 | 24,763 | 12,059 | 12,704 |
| 51 ~ 60 | 37,132 | 16,245 | 20,887 |
| 61 ~ 70 | 36,224 | 15,297 | 20,927 |
| 71 ~ 80 | 33,972 | 12,385 | 21,587 |
| 81 ~ 90 | 9,877 | 3,294 | 6,583 |
| $\geq$ 91 | 505 | 164 | 341 |

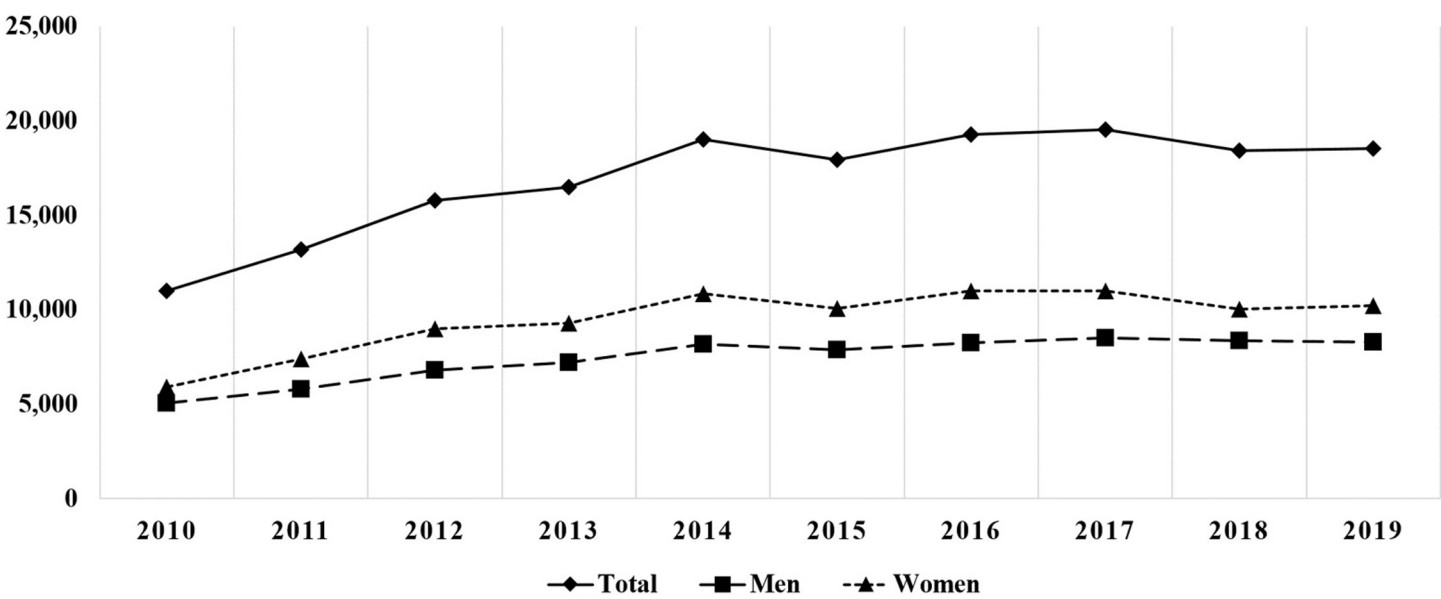

**Fig 2. The number of cases and sex distribution of IS from 2010 to 2019.**

study design, Kim et al. [17] reported an increase from 11.32 per 100,000 in 2007 to 13.35 per 100,000 in 2016 among people ≥ 20 years of age.

The incidence of IS has been increasing, mainly in older adults with multiple comorbidities and low immunity. This is due to the increase in the elderly population as well as increase in the number of patients with chronic diseases including DM, immunocompromised state, long-term steroid use, and drug abuse. This increase in IS incidence also reflects an increase in the early diagnosis of the disease due to the development of more advanced imaging modalities. The increase in invasive spinal procedures and surgeries has also contributed to the increasing incidence of IS, especially PS. A study analyzing data from HIRA reported that the number of patients who underwent surgery for lumbar spinal stenosis increased 2.54-fold, from 10,990 in 2003 to 27,942 in 2008. The number of patients undergoing outpatient epidural blocks also doubled from 40.8 per 1,000 in 2007 to 84.4 in 2015 [18, 19]. Kim et al. [7] reported

**Table 3. The number of patients who underwent surgical treatment for IS, stratified according to different surgical methods.**

| Year | Decompression | Corpectomy | Anterior fusion | Posterior fusion | Incision and drainage | Total |
|---|---|---|---|---|---|---|
| 2010 | 169 | 57 | 104 | 145 | 1,462 | 1,937 |
| 2011 | 190 | 59 | 113 | 148 | 1,492 | 2,002 |
| 2012 | 193 | 58 | 118 | 148 | 1,623 | 2,140 |
| 2013 | 151 | 39 | 93 | 128 | 1,650 | 2,061 |
| 2014 | 199 | 43 | 107 | 144 | 1,793 | 2,286 |
| 2015 | 184 | 54 | 109 | 168 | 1,757 | 2,272 |
| 2016 | 243 | 62 | 121 | 201 | 1,868 | 2,495 |
| 2017 | 246 | 36 | 127 | 197 | 2,114 | 2,720 |
| 2018 | 233 | 25 | 101 | 154 | 2,153 | 2,666 |
| 2019 | 252 | 26 | 132 | 184 | 2,041 | 2,635 |
| Total | 2,060 | 442 | 1,125 | 1,617 | 17,970 | 23,214 |
| P-value | 0.3473 | < 0.05 | < 0.05 | 0.4370 | < 0.05 | |

**Table 4. The prevalence of comorbidities in IS patients (Total: 169,244 patients).**

|  | No. of cases |
|---|---|
| DM | 93,199 (55.1%) |
| ESRD | 21,682 (12.8%) |
| RA | 46,146 (27.3%) |
| Autoimmune disease except RA | 3,890 (2.3%) |
| Liver cirrhosis | 6,037 (3.6%) |
| Chronic obstructive pulmonary disease | 25,742 (15.2%) |
| Immunodeficiency disease | 200 (0.1%) |
| Infective endocarditis* | 1,133 (0.7%) |

*It was calculated with a total of 153,333 PS patients as the denominator.

that 55% of patients diagnosed with IS had undergone at least one spinal procedure one month before the diagnosis.

Even considering these factors, the incidence rate per 100,000 people in this study increased from 22.90 in 2010 to 35.79 in 2019, which is significantly higher than the rate reported previously. While previous studies have excluded the incidence of IS cases after spinal procedures and surgeries, the present study did not exclude them entirely. Hence, all cases of IS were included in the present study. In addition, previous studies were conducted in the early and mid-2000s when the elderly population was relatively small and spinal procedures and surgeries were not very frequent [1, 4, 16].

IS can occur at any age. However, it has a bimodal distribution as it is more frequently diagnosed either in individuals under 20 or individuals 50–70 years of age [2]. In this study, individuals $\leq$ 20 years of age accounted for only 1.9% (3,264 patients) of the total number of patients, while individuals aged 50–70 years and elderly individuals $\geq$ 60 years of age accounted for 43.3% (73,356 patients) and 47.6% (80,578 patients) of the total number of patients, respectively. The immune system deteriorates with age. And comorbidities such as DM make the elderly population particularly vulnerable to infection. This contributes to the high incidence of IS in elderly patients [20]. In addition, degenerative spinal diseases that occur at older ages usually require invasive spinal procedures and surgeries. Thus, IS caused by procedures and surgeries is more commonly diagnosed in the elderly [18, 19, 21].

**Table 5. Proportion of health insurance type in patients treated for IS.**

| Year | No. of cases | | |
|---|---|---|---|
| | NHI | MAP | P-value |
| 2010 | 10,082 / 48,906,795 (2.1%) | 909 / 1,674,396 (5.4%) | < 0.05 |
| 2011 | 11,932 / 49,299,165 (2.4%) | 1,279 / 1,609,481 (7.9%) | < 0.05 |
| 2012 | 14,646 / 49,662,097 (2.9%) | 1,147 / 1,507,044 (7.6%) | < 0.05 |
| 2013 | 15,428 / 49,989,620 (3.1%) | 1,077 / 1,458,871 (7.4%) | < 0.05 |
| 2014 | 17,830 / 50,316,384 (3.5%) | 1,188 / 1,440,762 (8.2%) | < 0.05 |
| 2015 | 16,844 / 50,490,157 (3.3%) | 1,115 / 1,544,267 (7.2%) | < 0.05 |
| 2016 | 17,994 / 50,763,293 (3.5%) | 1,290 / 1,509,472 (8.5%) | < 0.05 |
| 2017 | 18,280 / 50,940,885 (3.6%) | 1,252 / 1,485,740 (8.4%) | < 0.05 |
| 2018 | 17,147 / 51,071,982 (3.4%) | 1,271 / 1,484,671 (8.6%) | < 0.05 |
| 2019 | 17,196 / 51,391,447 (3.3%) | 1,337 / 1,488,846 (9.0%) | < 0.05 |

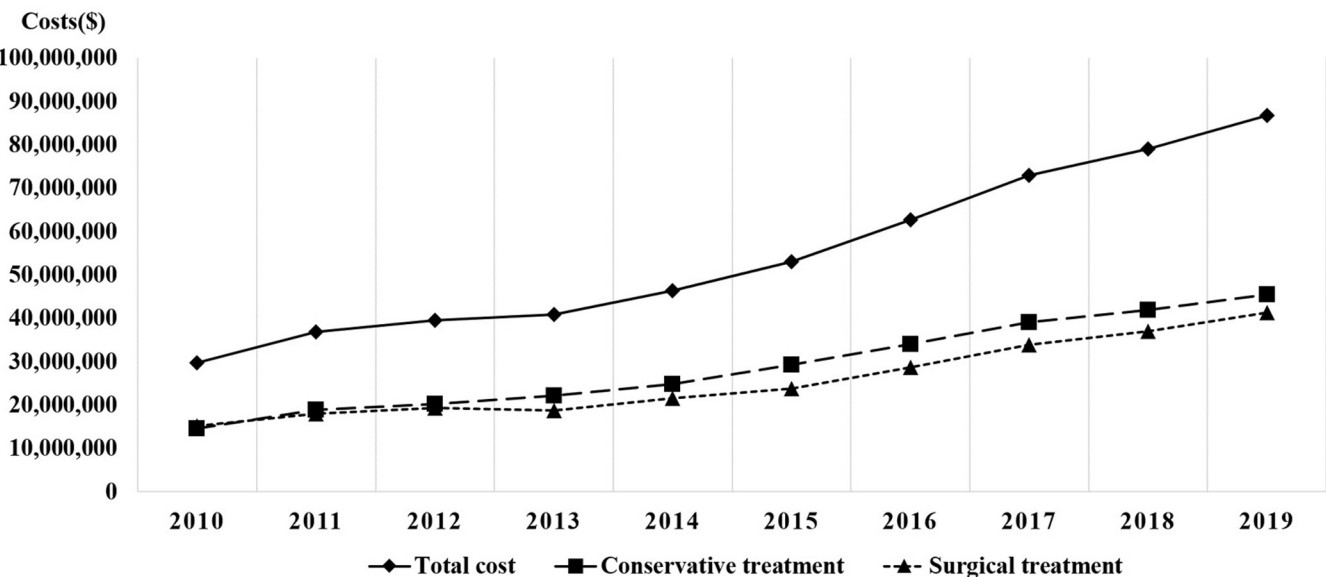

**Fig 3. The total healthcare costs for IS.**

Previous studies have reported a 1.5–2-fold higher incidence rate of IS in men compared with women. The cause of this difference has not yet been precisely identified. However, this could be due to the comparatively higher number of comorbidities in men compared to women [1, 2, 7]. In the present study, significantly more women were diagnosed with IS compared with men. Similarly, Kim et al. [17] reported that women had a higher incidence rate of IS than men analyzing the entire population of South Korea. A study investigating the epidemiology of spinal diseases in South Korea from 2012 to 2016 reported that women had a 1.4-fold higher incidence of spinal diseases than men, which could have contributed to the high incidence of IS as these patients are more likely to undergo spinal procedures and surgeries [22]. Higher rates of osteoporosis in women than in men could also be associated with higher rates of IS. According to a recent study, osteoporosis is a novel risk factor of infections and sepsis. Interleukin 7 and lipocalin 2-related interactions between bone metabolism and the immune system could partially explain the association of osteoporosis with an increased risk of infections and sepsis [23]. However, more research is required to investigate these sex differences and their underlying causes.

Over the past decade, the incidence of PS has increased, while that of TS has decreased. TS is mainly caused by hematogenous spread from the lungs or other primary lesions. Some tuberculosis bacteria can also be directly transmitted to the spine from organs around the spine, such as the lungs, kidneys, and digestive organs, hence, causing infection. Considering this mechanism of occurrence, the decrease in the incidence of TS could be associated with a reduction in the overall prevalence of tuberculosis. The prevalence of tuberculosis per 100,000 people had halved from 940 in 1965 to 443 in 1985 and subsequently, by one-quarter to 101 in 2015 in South Korea. However, the rate of tuberculosis infection remains high in the elderly population; thus, its diagnosis should not be neglected in this subset of patients [24].

The use of conservative treatment for IS has increased, while the use of surgical methods such as corpectomy and anterior fusion have decreased. This is because the clinicians took an active diagnostic approach as the incidence rate of IS increased. The development of advanced imaging modalities has supported these efforts, leading to an increase in the early diagnosis of IS. Moreover, the development of effective antibiotics against resistant bacteria has led to

sufficiently good results, even with conservative treatment alone [3, 10, 19, 25]. This could have reduced the number of cases of advanced IS requiring surgical treatments such as corpectomy and anterior fusion. This is consistent with the tendency to avoid highly invasive surgeries, as evidenced by the recent increasing trend in minimally invasive spine surgery [26, 27].

DM, advanced age, ESRD, RA, liver cirrhosis, cancer, and immunocompromised conditions are risk factors for IS. Uncontrolled DM is the most significant risk factor [2, 6, 28]. In this study, 55.1% of IS patients had DM, 27.3% had RA, and 12.8% had ESRD. However, this study did not examine whether these comorbidities were risk factors for IS.

In this study, 0.7% of patients had infective endocarditis, a rate lower than previous reports of 2–5% [1]. This could be due to the lack of routine echocardiography for patients diagnosed with IS, resulting in an underestimated incidence. Behmanesh et al. [29] reported incidence rates of infective endocarditis in patients undergoing and not undergoing routine transesophageal echocardiography of 33% and 3%, respectively, which is a significant difference. Infective endocarditis can cause serious complications such as cerebral infarction or aortic aneurysm. It is also associated with a very high mortality rate if not treated on time. Hence, routine echocardiography is recommended for patients diagnosed with IS.

Patients eligible for MAP developed more cases of IS compared with NHI enrollees; the total income differed between the two groups. Individuals with low incomes are more likely to have poor nutrition, making them more vulnerable to infectious diseases. In addition, such individuals also have limited access to medical institutions, which increases the time to diagnosis. Therefore, the index of suspicion for IS should be high in patients with low socioeconomic status complaining of back pain, fever, etc., and they should be tested accordingly [30, 31].

With the increase in the incidence of IS, the socioeconomic burden of managing this disease has also increased. The results of this study demonstrated that although the number of patients receiving surgical treatment was significantly lower than those receiving conservative treatments, the total healthcare costs between each group did not differ. This is because surgery usually requires higher medical expenses due to the addition of surgery costs, anesthesia fees, material costs, etc. If patients with suspected IS are diagnosed early by blood tests and imaging assessments, satisfactory results can be achieved via conservative treatment alone with antibiotics or anti-tuberculosis drugs. This would reduce the number of patients requiring surgical treatment, which will lead to a reduction in the overall socioeconomic burden caused by IS.

This study has several limitations. First, this study did not differentiate between infections due to spinal procedures, postoperative infections, or spontaneous infections. Considering the increasing trend of spinal procedures and surgeries, it would have been more helpful for clinical diagnosis if the incidences of IS had been differentiated according to its causes. Second, the exact causative pathogen type could not be identified in this study. The pathogen type is determined through culture; however, this information is often not accurately entered with the ICD-10 code. Finally, we could not analyze mortality due to IS. As IS is a life-threatening disease, studies investigating mortality rates are required; however, the direct causation of death is difficult to establish, especially in older patients with multiple comorbidities. Future studies are required to analyze this, along with risk factors that may affect mortality. Despite these limitations, the strengths of this study are that it identified the incidence rate, treatment trends, and characteristics of IS over 10 years in the entire population.

## Conclusions

This population-based cohort study demonstrated that the incidence rate of IS has increased in South Korea. The proportion of patients receiving conservative treatment has increased,

while that of patients who underwent surgical treatment has decreased. The socioeconomic burden of IS has increased rapidly.

## Author Contributions

**Conceptualization:** Hee Jung Son, Chang-Nam Kang.

**Data curation:** Hee Jung Son, Dong Hong Kim, Chang-Nam Kang.

**Formal analysis:** Hee Jung Son, Myongwhan Kim, Dong Hong Kim.

**Funding acquisition:** Hee Jung Son.

**Investigation:** Myongwhan Kim, Dong Hong Kim.

**Methodology:** Myongwhan Kim, Dong Hong Kim.

**Validation:** Hee Jung Son, Chang-Nam Kang.

**Writing – original draft:** Hee Jung Son.

**Writing – review & editing:** Hee Jung Son, Myongwhan Kim, Dong Hong Kim, Chang-Nam Kang.

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
