## [Decision Letter · Decision Letter 0]

10 Apr 2023

PONE-D-23-06492Incidence and Treatment Trends of Infectious Spondylodiscitis in South Korea: A Nationwide Population-Based StudyPLOS ONE

Dear Dr. Kang,

Thank you for submitting your manuscript to PLOS ONE. After careful consideration, we feel that it has merit but does not fully meet PLOS ONE’s publication criteria as it currently stands. Therefore, we invite you to submit a revised version of the manuscript that addresses the points raised during the review process.

ACADEMIC EDITOR: Please carefully read the reviewers' comments and respond adequately.

We look forward to receiving your revised manuscript.

Kind regards,

Hiroshi Hashizume, M.D., Ph.D.

Academic Editor

PLOS ONE

Journal Requirements:

Reviewers' comments:

Reviewer's Responses to Questions

**Comments to the Author**

1. Is the manuscript technically sound, and do the data support the conclusions?

Reviewer #1: Yes

Reviewer #2: Yes

2. Has the statistical analysis been performed appropriately and rigorously? 

Reviewer #1: Yes

Reviewer #2: Yes

3. Have the authors made all data underlying the findings in their manuscript fully available?

Reviewer #1: Yes

Reviewer #2: Yes

4. Is the manuscript presented in an intelligible fashion and written in standard English?

Reviewer #1: Yes

Reviewer #2: No

5. Review Comments to the Author

Reviewer #1: It is a useful data from big sample.

P4 line 68; Please explain about the abbreviation “TS”

P17 line 222；Provide citations for“several study”

Discussion

In the present results, women were more numerous and the age group with the highest number of respondents in their 50s. Can you please add further discussion on these matters?

Reviewer #2: I would like to thank the authors for submitting their wonderful work to the journal. It was my pleasure to have read the novel topic that investigated incidence and treatment trends of infectious spondylodiscitis (IS) in South Korea. The incidence rate of IS per 100,000 people in South Korea has increased 1.5-fold over the past 10 years, with Pyogenic spondylodiscitis increasing but tuberculosis spondylodiscitis decreasing. And conservative treatments increased, corpectomy and anterior fusion decreased among surgical treatments, and socioeconomic burden increased rapidly. I appreciate the scope/grandeur of such an important study work.

I have some comments for minor revise.

My specific comments on their work is noted below:

1; Introduction; “If bacteriological or histological confirmation is required, if there is intractable pain, if there is no improvement in symptoms despite appropriate antibiotics administration, if neurological paralysis occurs, or if there is spinal deformity, surgical treatment such as incision and drainage, decompression, corpectomy, and fusion may be performed.”

This sentence is not correct. Please look into by the English editor.

2; “ In recent years, the incidence of IS, especially PS, has been increasing due to the

increased numbers of older patients with chronic diseases, immunocompromised people, steroid use, drug abuse, invasive spinal procedures, and spinal surgeries.”

Please attach the references.

3; The author used female and male, sometimes women and men. Please unify the term.

6. PLOS authors have the option to publish the peer review history of their article (what does this mean?). If published, this will include your full peer review and any attached files.

Reviewer #1: No

Reviewer #2: **Yes: **Masatoshi Teraguchi

---

## [Decision Letter · Decision Letter 1]

13 Jun 2023

Incidence and Treatment Trends of Infectious Spondylodiscitis in South Korea: A Nationwide Population-Based Study

PONE-D-23-06492R1

Dear Dr. Kang,

We’re pleased to inform you that your manuscript has been judged scientifically suitable for publication and will be formally accepted for publication once it meets all outstanding technical requirements.

Kind regards,

Hiroshi Hashizume, M.D., Ph.D.

Academic Editor

PLOS ONE

Additional Editor Comments (optional):

All reviewers agree with the publication of your work. Congratulations!

Reviewers' comments:

Reviewer's Responses to Questions

**Comments to the Author**

1. If the authors have adequately addressed your comments raised in a previous round of review and you feel that this manuscript is now acceptable for publication, you may indicate that here to bypass the “Comments to the Author” section, enter your conflict of interest statement in the “Confidential to Editor” section, and submit your "Accept" recommendation.

Reviewer #1: All comments have been addressed

Reviewer #2: All comments have been addressed

2. Is the manuscript technically sound, and do the data support the conclusions?

Reviewer #1: Yes

Reviewer #2: Yes

3. Has the statistical analysis been performed appropriately and rigorously? 

Reviewer #1: Yes

Reviewer #2: Yes

4. Have the authors made all data underlying the findings in their manuscript fully available?

Reviewer #1: Yes

Reviewer #2: Yes

5. Is the manuscript presented in an intelligible fashion and written in standard English?

Reviewer #1: Yes

Reviewer #2: Yes

6. Review Comments to the Author

Reviewer #1: The author sincerely answered my questions. This paper is from a large sample and I think it will be useful.

Reviewer #2: Thank you for the revising article to be published. I feel that this study has been suitable for acceptance.

7. PLOS authors have the option to publish the peer review history of their article (what does this mean?). If published, this will include your full peer review and any attached files.

Reviewer #1: No

Reviewer #2: No

---

## [Editor Report · Acceptance letter]

19 Jun 2023

PONE-D-23-06492R1 

Incidence and Treatment Trends of Infectious Spondylodiscitis
in South Korea: A Nationwide Population-Based Study 

Dear Dr. Kang:

I'm pleased to inform you that your manuscript has been deemed suitable for publication in PLOS ONE. Congratulations! Your manuscript is now with our production department. 

Kind regards, 

on behalf of

Dr Hiroshi Hashizume 

Academic Editor

PLOS ONE